# FLOW TO BETTER: OFFLINE PREFERENCE-BASED RE-INFORCEMENT LEARNING VIA PREFERRED TRAJECTORY GENERATION

**Zhilong Zhang**[1,2][*]**Yihao Sun**[1][*]**Junyin Ye**[1,2]**, Tian-Shuo Liu**[1,2]**, Jiaji Zhang**[1]**, Yang Yu**[1,2][†]
[1]National Key Laboratory for Novel Software Technology, Nanjing University, China
&School of Artificial Intelligence, Nanjing University, China
[2]Polixir Technologies
{zhangzl,sunyh,yejy,liuts,zhangjj}@lamda.nju.edu.cn
yuy@nju.edu.cn

## ABSTRACT

Offline preference-based reinforcement learning (PbRL) offers an effective solution to overcome the challenges associated with designing rewards and the high costs of online interactions. Previous studies mainly focus on recovering rewards from preferences, followed by policy optimization with an off-the-shelf offline RL algorithm. However, given that preference labels in PbRL are inherently trajectory-based, accurately learning transition-wise rewards from such labels can be challenging, potentially leading to misguidance during subsequent offline RL training. To address this issue, we introduce our method named *Flow-to-Better (FTB)*, which leverages the pairwise preference relationship to guide a generative model in producing preferred trajectories, avoiding Temporal Difference (TD) learning with inaccurate rewards. Conditioning on a low-preference trajectory, *FTB* uses a diffusion model to generate a better one, achieving high-fidelity full-horizon trajectory improvement. During diffusion training, we propose a technique called *Preference Augmentation* to alleviate the problem of insufficient preference data. As a result, we surprisingly find that the model-generated trajectories not only exhibit increased preference and consistency with the real transition but also introduce elements of *novelty* and *diversity*, from which we can derive a desirable policy through imitation learning. Experimental results on several benchmarks demonstrate that FTB achieves a remarkable improvement compared to state-of-the-art offline PbRL methods. Furthermore, we show that FTB can also serve as an effective data augmentation method for offline RL. Our project's website can be found at https://github.com/Zzl35/flow-to-better.

## 1 INTRODUCTION

In reinforcement learning (RL), agents interact with an environment and receive feedback to learn a policy that maximizes cumulative rewards. This paradigm has demonstrated its efficacy in many domains, including games (Silver et al., 2017; Mnih et al., 2015), robotics (Levine et al., 2018), and large language model (Ouyang et al., 2022). However, crafting well-designed rewards that align with the task's objectives or human intentions presents a formidable challenge. First, we aim for the reward function to be densely informative, like transition-wise, to facilitate task learning. Secondly, the reward function must exhibit robustness to prevent policy exploitation loopholes that could result in unreasonably high rewards. Therefore, reward engineering necessitates a substantial foundation of prior knowledge and rigorous testing by human experts, making it extremely complex and even unfeasible in some cases.

Preference-based reinforcement learning (PbRL) tackles the challenge of designing reward functions by utilizing trajectory preferences (Akrour et al., 2011; 2012; Christiano et al., 2017). In this

---

[*]Equal contribution.
[†]Corresponding author.

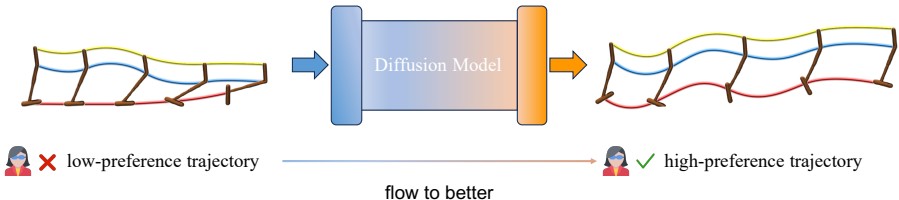

Figure 1: Illustration of the key idea of our method. Given a low-preference trajectory (left), the FTB model generates a higher-preference trajectory (right).

framework, agents are provided with preferences between pairs of trajectory segments, which are relative judgments labeled by humans. However, existing approaches still necessitate a substantial number of queries to human experts, making it difficult to scale online preference-based RL to various applications. For instance, even in simple video games or control tasks, existing methods still need millions of requests (Lee et al., 2021). Therefore, it becomes imperative to explore PbRL in an offline setting, where agents have access to a fixed offline dataset with preference labels, without concerns about safety and sample efficiency.

In offline PbRL, existing approaches (Shin & Brown, 2021; Kim et al., 2023) typically involve two steps: reward learning and offline RL. Specifically, agents employ the Bradley-Terry model (Bradley & Terry, 1952) in a supervised manner to learn a transition-wise reward function, followed by an off-the-shelf offline RL algorithm for policy optimization. However, preference labels in PbRL are trajectory-wise, posing challenges in recovering precise transition-wise rewards, which will significantly impact offline RL training and lead to undesirable policy performance.

To address this challenge, we aim to optimize policy behaviors directly at the trajectory level, as opposed to decomposing preferences into rewards and performing TD learning with these inaccurate rewards. Our innovative method, *Flow-to-Better (FTB)*, leverages the preference relationship to learn a generative model capable of improving a trajectory, thereby generating more high-preference trajectories. We employ a diffusion model to generate an improved full-horizon trajectory conditioned on the less preferred one, as illustrated in Figure 1. Additionally, we introduce *Preference Augmentation* to generate sufficient preference pairs for training the diffusion model. After diffusion model training, we apply a conditional generation process in an autoregressive manner to the original trajectories iteratively, effectively "flowing" low-preference trajectories to high-preference trajectories. With substantial high-preference trajectories generated by our model, we can extract a desirable policy through simple imitation learning. Overall, this approach introduces a new paradigm for offline PbRL to circumvent the pitfalls associated with inaccurate reward models.

We highlight the main contributions of our work below:

- We present a novel framework for offline PbRL, i.e., *Flow-to-Better*, which uses a trajectory diffuser to achieve trajectory optimization, avoiding TD learning with inaccurate rewards.
- We introduce *Preference Augmentation*, an innovative technique designed to alleviate the issue of insufficient preference labels in our approach.
- We demonstrate that FTB consistently outperforms previous offline PbRL methods across various complex locomotion and manipulation tasks.
- Our results show that the proposed trajectory diffuser in our method can also be used as an effective data augmentation method for existing offline RL approaches.

## 2 BACKGROUND

### 2.1 REINFORCEMENT LEARNING

We consider environments represented as a finite Markov decision process (MDP) (Sutton & Barto, 2018; Puterman, 2014), which is described by a tuple $(\mathcal{S}, \mathcal{A}, \mathcal{P}, \rho, \mathbf{r}, H)$, where $H$ is the length of an episode, $\mathcal{S}$ and $\mathcal{A}$ are the state and action space respectively, $\mathcal{P} = \{P_h(\cdot|s,a) : (s,a) \in \mathcal{S} \times \mathcal{A}\}_{h=1}^{H}$ and $\rho$ represent the dynamics and the initial state distribution, $\mathbf{r} = (r_1, \cdots, r_H)$ specifies the reward function. A policy $\pi_h : \mathcal{S} \to \mathcal{A}$ is a mapping from states to actions. The decision process runs as follows: an initial state $s_1$ is drawn from $\rho$ as the beginning, for each time step $h$, the agent observes

a state $s_h$, selects an action $a_h$ from the distribution $\pi_h(\cdot|s_h)$, and subsequently, the environment provides a reward $r_h(s_h, a_h)$ to the agent and transits to a new state $s_{h+1}$ according to $P_h(\cdot|s_h, a_h)$. The goal of the agent is to maximize the expected cumulative reward $\mathbb{E}[\sum_{h=1}^{H} r_h(s_h, a_h)]$.

**Offline preference-based reinforcement learning (Offline PbRL).** In contrast to online RL, offline PbRL (Shin et al., 2022; Kim et al., 2023) prohibits the agent from interacting with an environment, and there is no access to the reward function. Instead, the agent is provided with two sets of offline datasets. The first dataset, denoted as $\mathcal{D}^L = \{(\tau_m^0, \tau_m^1, y_m)\}_{m=1}^{M}$, comprises a limited number of trajectory pairs, each associated with a preference label $y_m \in \{0, 0.5, 1\}$, where $0$ implies that $\tau_m^0 \succ \tau_m^1$, $1$ implies $\tau_m^1 \succ \tau_m^0$, and $0.5$ implies $\tau_m^0 \sim \tau_m^1$. The second dataset, referred to as $\mathcal{D}^U = \{\tau_n\}_{n=1}^{N}$, comprises a large number of unlabeled trajectories. Most previous methods try to learn a reward function from $\mathcal{D}^L$ and label transitions in $\mathcal{D}^U$ for offline RL training.

## 2.2 DIFFUSION MODELS

Diffusion Models (DMs) are a sort of generative models that can model complex distributions and have achieved significant success in text to image (Dhariwal & Nichol, 2021; Rombach et al., 2022). The fundamental concept behind DMs is to progressively refine a noise-perturbed input to generate samples that closely resemble the target distribution. The forward process of DMs $q(\boldsymbol{x}_t|\boldsymbol{x}_{t-1}) = \mathcal{N}(\boldsymbol{x}_t; \sqrt{\alpha_t}\boldsymbol{x}_{t-1}, (1-\alpha_t)\boldsymbol{I})$ gradually diffuses from the target distribution to a Gaussian distribution in $T$ timesteps, while the reverse process $p_\theta(\boldsymbol{x}_{t-1}|\boldsymbol{x}_t) = \mathcal{N}(\boldsymbol{x}_{t-1}; \mu_\theta(\boldsymbol{x}_t, t), \Sigma_t)$ starts from a Gaussian distribution and iteratively denoises samples using a trained model, ultimately recovering the target distribution. The training objective of DMs is to minimize the negative log-likelihood $\mathbb{E}_{\boldsymbol{x} \sim q}[-\log p_\theta(\boldsymbol{x})]$, which can be simplified using the variational lower bound in Equation 1.

$$\mathcal{L}_{\text{VLB}}(\theta) := \mathbb{E}_{t \sim [1,T], \boldsymbol{x} \sim q, \epsilon \sim \mathcal{N}(\boldsymbol{0}, \boldsymbol{I})}[\|\epsilon - \epsilon_\theta(\boldsymbol{x}_t, t)\|^2] \tag{1}$$

**Guidance Diffusion Models.** Guidance DMs model the conditional distribution $q(\boldsymbol{x}|y)$. This process can be understood through a score function $\nabla \log p(\boldsymbol{x}_t|y) = \nabla \log p(\boldsymbol{x}_t) + \nabla \log p(y|\boldsymbol{x}_t)$ (Song et al., 2020). Intuitively, we need to simultaneously model both the unconditional score $\nabla \log p(\boldsymbol{x}_t)$ and classifier guidance score $\nabla \log p(y|\boldsymbol{x}_t)$. Currently, methods for guidance DMs primarily fall into two categories: classifier-guidance (Dhariwal & Nichol, 2021) and classifier-free (Nichol et al., 2022). The former entails training an additional classifier guidance estimator, denoted as $f_\phi(y|\boldsymbol{x}_t)$, alongside the DMs. During generation, it samples perturbed noise $\bar{\epsilon}_\theta(\boldsymbol{x}_t, t) := \epsilon_\theta(\boldsymbol{x}_t, t) - \omega\sqrt{1-\bar{\alpha}_t}\nabla \log f_\phi(y|\boldsymbol{x}_t)$. On the other hand, the latter approach involves training both conditional models $\epsilon_\theta(\boldsymbol{x}_t, y, t)$ and unconditional models $\epsilon_\theta(\boldsymbol{x}_t, t)$ throughout the DM training process. During generation, it samples perturbed noise $\bar{\epsilon}_\theta(\boldsymbol{x}_t, t) := \epsilon_\theta(\boldsymbol{x}_t, t) + \omega(\epsilon_\theta(\boldsymbol{x}_t, y, t) - \epsilon_\theta(\boldsymbol{x}_t, t))$. Here, $\omega$ serves as a hyperparameter to adjust the degree of conditional guidance.

## 3 METHOD

This section introduces our *Flow-to-Better (FTB)* method, which optimizes policies at the trajectory level without TD learning under inaccurate learned rewards. We treat offline preference-based reinforcement learning as a conditional generation task, where a diffusion model generates higher-preference trajectories based on inferior trajectories (Section 3.1). Given that a significant portion of the offline dataset lacks labels, we introduce a method called *Preference Augmentation*, which can provide more preference pairs for diffusion training. (Section 3.2). Furthermore, we explain how to derive a deployable policy (Section 3.3) and provide a full procedure of our method (Section 3.4).

## 3.1 OFFLINE PREFERENCE-BASED RL AS A CONDITIONAL GENERATION TASK

In offline PbRL, we have pairs of trajectories with preference labels, enabling agents to learn from relative judgments—progressing from an inferior trajectory (low preference) to a superior one (high preference). To accomplish this, we formulate the problem as a conditional generation task, as depicted in Equation 2. In this task, our goal is to maximize the log-likelihood of the conditional distribution of generating improved trajectories. Here, the condition is the less preferred trajectory in human judgment.

$$\min_\theta \mathbb{E}_{(\tau^1 \succeq \tau^0) \sim \mathcal{D}^L} \left[ -\log p_\theta(\tau^1|\tau^0) \right] \tag{2}$$

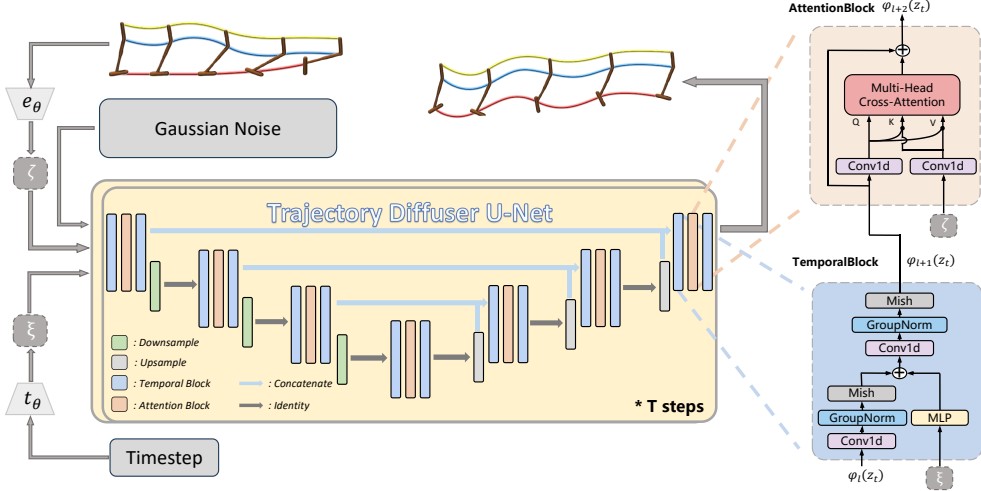

Figure 2: The architecture of trajectory diffuser, which is a classifier-free diffusion model.

Note that this generation task is highly challenging, as full-horizon trajectories typically have high dimensions. Furthermore, there exist multiple diverse and distinct trajectories that outperform the conditioned trajectory, leading to a multi-modal target distribution. Therefore, we employ a classifier-free diffusion model, *Trajectory Diffuser*, as the central implementation of our generative model. The training loss is defined as Equation 3:

$$\mathcal{L}^{\text{TrajDiffuser}}(\theta) := \mathbb{E}_{t,\epsilon\sim\mathcal{N}(\mathbf{0},\mathbf{I}),(\tau^1\succeq\tau^0)\sim\mathcal{D}^L,\beta\sim\text{Bern}(p)}[\left\|\epsilon - \epsilon_\theta(\tau_t^1, (1-\beta)\tau^0 + \beta\varnothing, t)\right\|^2], \quad (3)$$

where $t$ is the diffusion timestep sampled from $t \sim \mathcal{U}\{1, ..., T\}$, $\tau_0$ is the low-preference conditional trajectory, and $\tau_t^1$ is the high-preference target trajectory in the $t$ diffusion timestep. Note that with probability $p$, we ignore the conditional information, and $\varnothing$ is a dummy value taking the place of condition $\tau^0$.

In Figure 2, we illustrate the architecture of the trajectory diffuser in FTB, which includes several encoders. Notably, we introduce a trajectory encoder denoted as $e_\theta$ designed to embed the reference trajectory. This encoder employs a one-dimensional convolutional network to map a trajectory $\tau \in \mathbb{R}^{H \times (d_\mathcal{S} + d_\mathcal{A})}$ to a representation $\zeta \in \mathbb{R}^{H \times d_\tau}$, where $H$ represents the trajectory horizon. This encoder compresses the dimensions of the state ($d_\mathcal{S}$) and action ($d_\mathcal{A}$) into $d_\tau$ channels. Subsequently, $\zeta$ is passed through an attention block with a U-Net architecture, undergoing multi-head cross-attention. This produces intermediate representations $\varphi(z_t) \in \mathbb{R}^{H \times d_\tau}$ in preceding layers. Following a similar approach to (Ho et al., 2020), we utilize a timestep encoder denoted as $t_\theta$ to generate representations $\xi \in \mathbb{R}^{d_\tau}$. These derived representations $\xi$ are integrated into the temporal block, serving as conditions alongside the intermediate representation $\varphi_t(z_t)$.

## 3.2 PREFERENCE AUGMENTATION

Although the method in Section 3.1 trains a generator that improves trajectories, there are still the following issues in practice: 1) The aforementioned method requires preference labels for each pair of trajectories to conduct training, meaning that it can only use the dataset $\mathcal{D}^L$ in which each pair of trajectories has a preference label while wasting the unlabeled dataset $\mathcal{D}^U$. 2) The performance differences between trajectory pairs are misaligned. Some trajectory pairs may perform similarly, while others exhibit substantial differences. This misalignment also challenges the model's learning.

To tackle these issues, we propose *Preference Augmentation*, a method that uses a learned preference model, $s_\psi$, to assign preferences to unlabeled trajectories and organize them into performance-based blocks. This preference model takes a trajectory as input and is trained on the supervised dataset $\mathcal{D}^L$ following the Bradley-Terry model (Bradley & Terry, 1952) as Equation 4.

$$P[\tau^1 \succ \tau^0; \psi] = \frac{\exp s_\psi(\tau^1)}{\exp s_\psi(\tau^0) + \exp s_\psi(\tau^1)} \quad (4)$$

**Remark 3.1** *Equation 4 is similar to the loss function used for reward learning in previous methods. However, the significant difference is that it models trajectory-wise scores $s(\tau)$ rather than transition-wise rewards $r(s, a)$, reducing learning complexity. Extensive experiments in Section 4.2 have revealed that preference models can precisely forecast ground truths than reward models.*

Then, we employ this preference model to score each trajectory in the unlabeled dataset $\mathcal{D}^U$. Finally, we cluster the trajectories into $K$ blocks based on their scores, resulting in several blocks with scores ascending in order $B_1 \prec \cdots \prec B_K$. The detailed process is outlined in Algorithm 2. We reformulate the conditional generation task as Equation 5, where the pair of trajectories is randomly sampled from neighboring blocks, and the worse one is employed as the condition while the better one serves as the target. Not only does it provide more pairs of trajectories for training but also ensures that the performance gap between each pair of trajectories is relatively consistent. The specific training process is shown as Algorithm 3.

$$\min_{\theta} \frac{1}{K-1} \sum_{k=1}^{K-1} \mathop{\mathbb{E}}_{\substack{t,\epsilon,\beta\sim\mathrm{Bern}(p) \\ \tau^0\sim B_k, \tau^1\sim B_{k+1}}} [\|\epsilon - \epsilon_\theta(\tau_t^1, (1-\beta)\tau^0 + \beta\varnothing, t)\|^2] \tag{5}$$

### 3.3 POLICY EXTRACTION

A well-trained generative model leans to improve complete trajectories, which inspires us to generate trajectories that approach or even surpass the optimal one in the dataset. However, our ultimate goal is to derive a policy that can be deployed in the environment. To this end, we adopt a simple yet practical policy extraction algorithm as shown in Algorithm 4. Specifically, we first select the top $k$ trajectories (according to preference scores) from the dataset $\mathcal{D}^U$ and use them as inputs for the diffusion model to improve. Then, we iteratively refine them by applying this process $t_{\mathrm{flow}}$ times, where $t_{\mathrm{flow}}$ is a self-adaptive hyperparameter. In the end, we obtain a batch of trajectories with high performance so that it is natural to employ imitation learning to derive the policy. It is worth emphasizing that, unlike most previous algorithms that use generative models for planning, this approach allows us to leverage the powerful capabilities of the generative model while avoiding the resource-intensive issues associated with using the generative model during inference.

### 3.4 FULL PROCEDURE

In summary, our method can be divided into three parts: 1) **Preference Augmentation**. This involves training a preference score model using the labeled dataset $\mathcal{D}^L$ and arranging the unlabeled dataset $\mathcal{D}^U$ into a sequence of ascending blocks as $B_1 \prec \cdots \prec B_K$. 2) **Generative Model Training**. Training a classifier-free diffusion model capable of generating better trajectories than the given ones. 3) **Policy Extraction**. Applying imitation learning to derive the policy from the trajectories generated through iterative improvement. For the complete algorithm, please refer to Algorithm 1.

---

**Algorithm 1** Flow to Better

**Require:** labeled data $\mathcal{D}^L = \{(\tau_m^1 \succeq \tau_m^0)\}_{m=1}^M$, unlabeled data $\mathcal{D}^U = \{\tau_n\}_{n=1}^N$,
             block number $K$, number of trajectories for improvement $k$, flow step $t_{\mathrm{flow}}$.
1: $s_\psi, \{B_k\}_{k=1}^K$ = Preference Augmentation($\mathcal{D}^L, \mathcal{D}^U, K$) (Algorithm 2).
2: $p_\theta$ = Generative Model Training($\{B_k\}_{k=1}^K$) (Algorithm 3).
3: $\pi_\phi$ = Policy Extraction($\mathcal{D}^U, s_\psi, p_\theta, k, t_{\mathrm{flow}}$) (Algorithm 4).
**Output:** policy $\pi_\phi$.

---

## 4 EXPERIMENTS

In this section, we conduct experiments to answer the following questions:

- How well does FTB perform compared with other offline PbRL baselines? (Section 4.1)
- Is preference learning easier than reward learning? (Section 4.2)
- What about the quality of trajectories generated by FTB? Are they better than the original datasets? (Section 4.3)
- Can FTB serve as a data augmentation method for offline RL? (Section 4.4)

## 4.1 Benchmark Results

**Benchmarks.** To make a comprehensive evaluation of our methods, we choose several continuous control tasks from two benchmarks: D4RL (Fu et al., 2020), and MetaWorld (Yu et al., 2020), in which D4RL can represent locomotion tasks while MetaWorld involves manipulation tasks. As for the collection of preferences, we use synthetic preferences from scripted teachers, which generates preferences based on ground-truth reward $r$ as follows: $y = i$, where $i = \arg\max_{i \in \{0,1\}} \sum_{h=1}^{H} r(s_h^i, a_h^i)$. We collect **15 preference labels** for locomotion tasks (D4RL) and **30 preference labels** for manipulation tasks (MetaWorld), significantly fewer than those used in previous work (Kim et al., 2023; Hejna & Sadigh, 2023). More details about these benchmarks and the collection of preferences are shown in Appendix A. Furthermore, we also conduct ablation experiments, including analyses of preference sources, model architecture, hyperparameters, and the dataset distribution (Qin et al., 2022). The results are available in Appendix C.

**Baselines.** We compare FTB with various algorithms, including three widely-used offline RL methods: 1) **10%BC**: imitating the top 10% high-performance samples; 2) **TD3+BC** (Fujimoto & Gu, 2021): adopting a BC constraint when optimizing policy; 3) **IQL** (Kostrikov et al., 2022): using expectile regression for Q-learning; and three offline PbRL methods: 1) **IQL+$r_\psi$**: performing IQL with a learned reward function; 2) **OPRL** (Shin et al., 2022): performing IQL with ensemble-diversed-based reward functions; 3) **PT** (Kim et al., 2023): performing IQL with a reward function learned by preference transformer; 4) **IPL** (Hejna & Sadigh, 2023): leveraging a soft-Bellman operator which computes the mapping from $\mathcal{Q}$-function to rewards to avoid reward function learning.

Table 1 shows the results based on scripted teacher preferences [*]. FTB consistently outperforms the other offline PbRL algorithms by a large margin across most tasks, indicating that direct optimization on trajectories is a better way for offline PbRL than learning proxy rewards. Remarkably, FTB is even comparable to these offline RL algorithms, suggesting that our approach can yield competitive policies without the need for a meticulously crafted reward function.

Table 1: Performance with preferences from scripted teachers, averaged over 5 random seeds. Among the offline PbRL methods, we bold the highest score for each task. Among all the methods, we mark the highest score with "*" for each task.

| Task Name | 10%BC | TD3+BC | IQL | IQL+$r_\psi$ | OPRL | PT | IPL | FTB (Ours) |
|---|---|---|---|---|---|---|---|---|
| halfcheetah-medium-replay-v2 | 23.6 | 48.1 | 48.3* | 36.0 | 38.3 | **41.1** | 36.5 | 38.4±1.3 |
| halfcheetah-medium-expert-v2 | 90.1 | 90.8 | 94.7* | 85.6 | 82.9 | **85.8** | 76.7 | 85.2±0.7 |
| hopper-medium-replay-v2 | 70.4 | 64.4 | 97.4* | 20.8 | 86.2 | 31.4 | 21.2 | **89.6±4.9** |
| hopper-medium-expert-v2 | 111.2* | 101.2 | 107.4 | 88.5 | 95.7 | 77.8 | 91.6 | **111.1±2.0*** |
| walker2d-medium-replay-v2 | 54.4 | 85.6* | 82.2 | 75.7 | 64.0 | **79.6** | 8.8 | 79.1±1.4 |
| walker2d-medium-expert-v2 | 108.7 | 110.0 | 111.7* | **110.0** | 109.6 | 109.4 | 78.7 | 109.3±0.3 |
| **D4RL-Average** | 76.4 | 83.4 | 90.3* | 69.4 | 79.5 | 70.9 | 52.3 | **85.5±1.8** |
| assembly-v2 | 0.02 | 0.00 | 0.17* | 0.00 | 0.00 | 0.00 | 0.01 | **0.02±0.00** |
| button-press-v2 | 0.70 | 0.61 | 0.70 | 0.23 | 0.57 | 0.16 | 0.11 | **1.00±0.00*** |
| drawer-open-v2 | 0.77 | 1.00* | 1.00* | 0.26 | 0.63 | 0.85 | 0.45 | **1.00±0.00*** |
| plate-slide-v2 | 0.43 | 0.58 | 0.62* | 0.00 | 0.44 | 0.45 | 0.38 | **0.51±0.08** |
| sweep-into-v2 | 0.07 | 0.63 | 0.57 | 0.00 | 0.21 | 0.18 | 0.14 | **0.97±0.02*** |
| **MetaWorld-Average** | 0.40 | 0.56 | 0.61 | 0.10 | 0.37 | 0.33 | 0.22 | **0.7±0.02*** |

## 4.2 Comparison of Reward Learning and Preference Learning

We argue that learning an accurate reward is impractical in the offline PbRL setting, especially when preference labels are scarce. In contrast, we can easily derive a faithful preference model. To substantiate this claim, we evaluate both the learned reward model and the learned preference model within the scripted teacher setting. Here, we can directly use real returns as ground truths for preferences. Figure 3 presents a comparison between predictions and ground truths of rewards and preferences in two tasks. Notably, learned preferences exhibit a significantly stronger correlation with their ground truths compared to learned rewards. This suggests that modeling trajectory-wise

---

[*]The results of offline RL baselines are sourced from https://github.com/tinkoff-ai/CORL. The results of offline PbRL baselines are reproduced by official codes, and we report the score of the last epoch over 5 seeds.

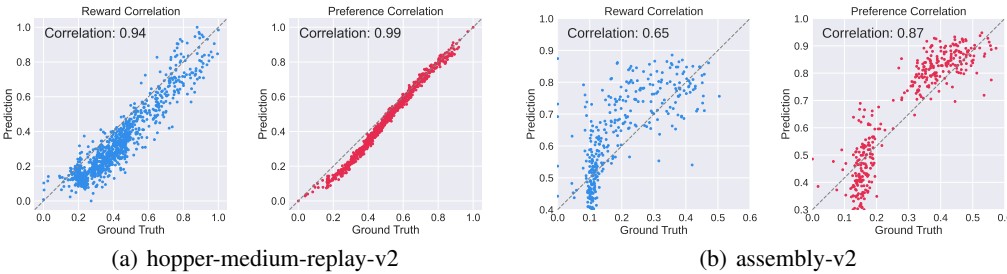

Figure 3: Illustration of the correlation between learned rewards/preferences and their ground truths. During training, hopper-medium-replay has 15 preference labels and assembly-v2 has 30 preference labels. We sample data from unlabeled datasets for evaluation.

preferences is a more tractable approach in the offline PbRL setting. We provide more results on other tasks in Appendix C.5, which is in accord with this conclusion without exception.

## 4.3 GENERATED TRAJECTORY ANALYSIS

In this section, our primary focus is on the quality of trajectories generated through the flow-to-better process. Specifically, we evaluate three critical aspects: 1) **Improvement**: whether these generated trajectories exhibit higher preferences compared to the original trajectories within the dataset. 2) **Validity**: whether they are faithful to the underlying real transition function. 3) **Generalization**: whether the generated trajectories are novel and diverse.

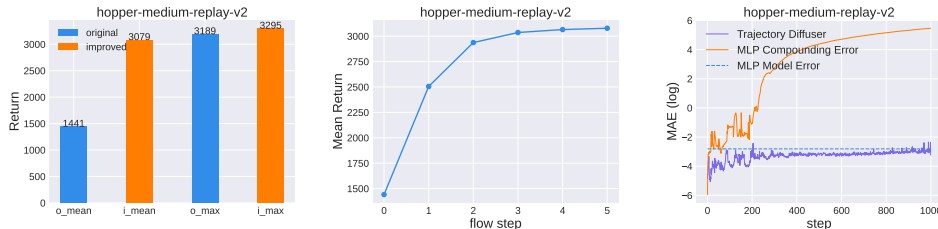

Figure 4: Illustration of performance before and after improvement (left), performance with different flow steps (center), and the dynamics error of generated trajectories (right).

**Improvement:** We expect that FTB can generate better trajectories. To verify this, we compare the original trajectories (top $k$ trajectories from the dataset) and the generated trajectories in terms of real returns [*]. Note that in the scripted teacher setting, higher returns correspond to higher preferences. As illustrated in Figure 4 (left), the generated trajectories exhibit a higher average return (labeled as "i_mean") than the original trajectories (labeled as "o_mean"). Meanwhile, the maximum return (labeled as "i_max") in generated trajectories also surpasses that of the original trajectories, indicating that FTB indeed extrapolates better samples. Furthermore, Figure 4 (center) shows more details during the FTB process. Here, we observe that as the number of flows increases, there is a corresponding improvement in overall performance.

**Validity:** We expect our generated trajectories to be faithful to the underlying real transition function. In order to assess the validity of generated trajectories, we obtain real transition $(\hat{s}_h, \hat{a}_h, s_{h+1})$ by means of the oracle dynamics model for each $(\hat{s}_h, \hat{a}_h)$ in generated trajectories and compute mean absolute error $\frac{1}{|\mathcal{S}|} \|s_{h+1} - \hat{s}_{h+1}\|_1$. To provide a comprehensive view of error magnitudes, we additionally train an MLP-based model to predict these transitions and calculate their errors. Figure 4 (right) illustrates model errors at different steps within these trajectories, alongside the average error of the MLP model (indicated by dashed lines) and the compounding error of the MLP model (indicated by the orange line). Notably, the overall model error in the generated trajectories remains consistently low and uniformly distributed across different steps, highlighting the diffusion model's impressive capability to generate full-horizon trajectories.

---

[*]For generated trajectories, we use the ground-truth reward function to compute each transition in them and get the real return.

**Generalization:** We expect our generated trajectories to be diverse and novel. To quantify generalization, We compute the L2 distance distance between the generated trajectories and the dataset as $\mathcal{L}_2(\hat{\tau}) = \min_{\tau \in \mathcal{D}^U} \|\hat{\tau} - \tau\|_2$. We plot a joint scatter plot about L2 distance and dynamic error in the form of mean absolute error (MAE). We also compare our method with Gaussian augmented $\mathcal{N}(0, 0.1)$ refer to Lu et al. (2023). As shown in Figure 5, the mean and variance of the L2 distance distribution for the samples generated by our model are significantly higher than those of the Gaussian perturbation samples. This indicates that our model is not merely replicating samples from the dataset but rather generating samples that exhibit diversity and novelty. In summary, our approach is capable of generating diverse and novel samples while maintaining sufficiently low dynamic errors.

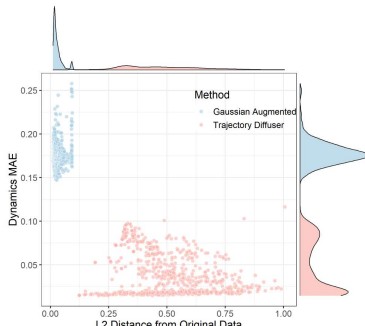

Figure 5: Comparing L2 distance from training data and dynamics accuracy under FTB generation and Gaussian augmentations.

## 4.4 APPLICATION IN OFFLINE RL

Since the trajectory diffuser in our method can generate synthetic data, it has the advantage of being able to serve as a data augmentation method for standard offline RL. We notice that there has been some work (Lu et al., 2023) using the diffusion model for data augmentation. However, all of them can only generate synthetic samples in accordance with the distribution of the original dataset. We argue that data augmentation for offline RL also needs samples with better performance since this can explicitly improve the performance bottleneck of an algorithm. With our proposed trajectory diffuser, we can supplement more performant samples.

To verify the effectiveness of FTB as a data augmentation method, we evaluate it in combination with 2 widely-used SOTA offline RL algorithms: TD3+BC (Fujimoto & Gu, 2021) and IQL (Kostrikov et al., 2022) on various Gym-Mujoco locomotion tasks in the D4RL benchmark. For comparison, we also include another diffusion-based data augmentation method, SYNTHER (Lu et al., 2023). We refer to more details about the training and augmentation process in Appendix B.5. The final performance is shown in Table 2. We notice significant improvements across these tasks with the data augmentation via FTB while the advantage of SYNTHER is less obvious compared with ours.

Table 2: Evaluation of our method as a data augmentation method for different offline RL algorithms, averaged over 5 seeds. We highlight the highest average score for each group of algorithms.

| Task Name | TD3+BC | | | IQL | | |
|---|---|---|---|---|---|---|
| | Original | SYNTHER | TrajDiffuser (Ours) | Original | SYNTHER | TrajDiffuser (Ours) |
| halfcheetah-medium-replay-v2 | 44.8 | 45.4 | 44.7 | 44.5 | 46.7 | 44.1 |
| halfcheetah-medium-expert-v2 | 90.8 | 85.9 | 94.7 | 94.7 | 93.6 | 92.6 |
| hopper-medium-replay-v2 | 64.4 | 54.0 | 80.2 | 97.4 | 102.8 | 100.4 |
| hopper-medium-expert-v2 | 101.2 | 102.5 | 101.5 | 107.4 | 97.5 | 111.2 |
| walker2d-medium-replay-v2 | 85.6 | 91.9 | 93.6 | 82.2 | 90.2 | 81.6 |
| walker2d-medium-expert-v2 | 110.0 | 110.1 | 110.2 | 111.7 | 111.8 | 111.3 |
| **Average** | 82.8 | 81.6 | **87.5** | 89.7 | **90.4** | **90.2** |

## 5 RELATED WORK

**Preference-based Reinforcement Learning.** Preference-based reinforcement learning (PbRL) is a promising paradigm that leverages human preferences to train RL agents, eliminating the need for reward engineering. Christiano et al. (2017) pioneer deep neural networks for online PbRL, which achieves remarkable success in solving complex control tasks. Building on this foundation, Ibarz et al. (2018) enhance the sample efficiency of this method by incorporating expert demonstrations. Subsequently, PEBBLE (Lee et al., 2021) introduces a novel approach by combining off-policy learning and pre-training, resulting in substantial improvements in feedback efficiency.

With the recent advancements in offline reinforcement learning (Fujimoto & Gu, 2021; Kostrikov et al., 2022; Jin et al., 2022; Sun et al., 2023; Luo et al., 2024), there has been a growing interest in the offline preference-based RL (offline PbRL) setting, which holds significant practical promise

for real-world applications by eliminating the need for an interactive environment or a predefined reward function. OPAL (Shin & Brown, 2021) is the first offline PbRL algorithm combining a previous online PbRL method and an off-the-shelf offline RL algorithm. PT (Kim et al., 2023) adopts a transformer architecture to design reward models capable of generating non-Markovian rewards. These approaches all involve reward learning and policy evaluation with imperfect reward models. To avoid the misdirection of inaccurate rewards, OPPO (Kang et al., 2023) and IPL (Hejna & Sadigh, 2023) construct algorithms beyond reward learning. However, in contrast to ours, OPPO achieves this by introducing an offline hindsight information matching objective for optimizing a contextual policy, while IPL adopts a soft-Bellman operator which computes the mapping from $Q$-functions to rewards under a fixed policy to avoid learning rewards.

**Generative Models in RL.** Generative models have demonstrated exceptional capabilities in generating text and images (OpenAI, 2023; Saharia et al., 2022; Nichol et al., 2022; Nichol & Dhariwal, 2021). Inspired by their remarkable success in these domains, an increasing number of researchers are applying generative models to RL. According to their structures, existing methods can be divided into two categories: Transformer-based and Diffusion-based. Within the Transformer-based methods, notable examples include Decision Transformer (Chen et al., 2021) and Trajectory Transformer (Janner et al., 2021), which approach Markov decision processes as a sequence modeling problem, with the goal being to produce a sequence of actions that leads to a sequence of high rewards. Moreover, Algorithm Distillation (Laskin et al., 2022) offers an intriguing approach by encoding past trajectories as contextual information and distilling the learning process of algorithms to rapid adaptation to novel tasks. In a similar vein, Agentic Transformer (Liu & Abbeel, 2023) leverages sorted historical trajectories as prompts to distill latent insights on trajectory improvement, thereby guiding the selection of more optimal actions, which provides a practical guarantee for achieving such improvements for our method. Within the Diffusion-based methods, Diffuser (Janner et al., 2022) uses a diffusion model as a trajectory generator and learns a separate return model to guide the reverse diffusion process toward samples of high-return trajectories. The consequent work, Decision Diffuser (Ajay et al., 2023) introduces conditional diffusion with reward or constraint guidance for decision-making tasks, further enhancing Diffuser's performance. In a different approach, SYNTHER (Lu et al., 2023) extends the application of diffusion models to data augmentation for RL. Distinguishing itself from these methods, our work explores the untapped potential of diffusion models in the realm of offline PbRL. To the best of our knowledge, FTB stands as the pioneering method capable of achieving end-to-end trajectory improvement in this field.

## 6 CONCLUSION

**Summary.** In this paper, we propose a novel diffusion-based framework for offline PbRL, i.e., Flow-to-Better (FTB). FTB treats the offline PbRL problem as a conditional generation task and uses a diffusion model to generate high-preference trajectories when given low-preference trajectories as conditions, achieving direct trajectory-wise improvement without TD learning under inaccurate learned rewards. With large amounts of high-preference trajectories generated by FTB, we can use imitation learning to derive a desirable policy. For better preference efficiency, we additionally propose the Preference Augmentation method to alleviate the scarce preference labels during learning. Empirically, we find that the model-generated trajectories not only exhibit increased preference and consistency with the real transition but also introduce elements of novelty and diversity. Experimental results on several benchmarks demonstrate that FTB achieves a remarkable improvement compared to previous offline PbRL methods. Furthermore, we show that FTB can also be used as an effective data augmentation method for offline RL approaches.

**Limitations and Future Work.** Compared to previous methods, FTB has a higher demand for computational resources. As shown in Appendix B.1, FTB requires 1-2 days to be trained on an RTX 4090 and needs more GPU memory, since the computation of the attention-based diffusion model grows quadratically with the task's horizon. Therefore, when the task horizon is excessively long, the training duration of FTB will significantly increase. However, drawing inspiration from Rombach et al. (2022), future work could look into training diffusion models in a latent space rather than the original observation space, which may effectively reduce the computational resource overhead.

ACKNOWLEDGMENTS

This work was supported by the National Science Foundation of China (61921006).

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

## A    TASK DETAILS

In this section, we describe the details of tasks from D4RL (Fu et al., 2020) and MetaWorld (Yu et al., 2020) as well as preference collection.

**Locomotion Tasks**. We choose several locomotion tasks from the D4RL benchmark. The goal of these tasks is to control the robots to move forward while minimizing the energy cost for safe behaviors. We select three environments (*halfcheetah*, *hopper* and *walker2d*) and two dataset types (*medium-replay* and *medium-expert*) per environment. We use the normalized score provided by D4RL as the evaluation metric. Additionally, we also consider the NeoRL (Qin et al., 2022) benchmark for locomotion tasks in Appendix C.3, which has a narrower data distribution.

**Manipulation Tasks** MetaWorld is a benchmark with simulated manipulation tasks with everyday objects, all of which are contained in a shared, table-top environment with a simulated Sawyer arm. We choose five tasks from this benchmark: 1) *assembly*: pick up a nut and place it onto a peg; 2) *button-press*: press a button; 3) *drawer-open*: open a drawer; 4) *plate-slide*: slide a plate into a cabinet; 5) *sweep-into*: sweep a puck into a hole. Offline datasets for these five tasks are constructed as the same as in IPL (Hejna & Sadigh, 2023): collect 100 trajectories of expert data; collect 100 trajectories of sub-optimal data and 100 trajectories of even more sub-optimal data; and collect 100 trajectories with uniform random actions. We use the success rate for a specific task as the evaluation metric.

**Preference Collection**. We generate preferences based on ground-truth reward $r$ as follows: $y = i$, where $i = \arg\max_{i \in \{0,1\}} \sum_{h=1}^{H} r(s_h^i, a_h^i)$. We collect 15 preference labels for locomotion tasks (D4RL) and 30 preference labels for manipulation tasks (MetaWorld), significantly fewer than those used in previous work (Kim et al., 2023; Hejna & Sadigh, 2023). We also consider the preferences given by real humans and show the results in Appendix C.4.

## B    IMPLEMENTATION DETAILS

### B.1    COMPUTATIONAL RESOURCE

We train FTB on an RTX 4090, with approximately 2 days required for one run. Specifically, Preference Augmentation takes only around 2 minutes, while Generative Model Training consumes approximately 40 hours. The time needed for Policy Extraction varies depending on the specific task, ranging from 30 minutes to 2 hours. For each task, we run our experiments on 5 seeds. We list detailed computational consumptions in the following Table.

Table 3: Computational consumption of different algorithms.

| Training time / GPU memory | PT / OPRL | FTB |
|---|---|---|
| D4RL/NeoRL | 3h / 1Gb | 36h / 23Gb |
| MetaWorld | 3h / 1Gb | 15h / 8Gb |

### B.2    PREFERENCE AUGMENTATION

The key component in Preference Augmentation is the score model $s_\psi(\tau)$, which is used to assign scores to each trajectory in the unlabeled dataset. Recall that $s_\psi(\tau)$ takes a trajectory as input and outputs a score. To achieve this, we can use any temporal model like Transformer or RNN to implement it. Alternatively, we can simply reuse the MLP-based reward model in previous methods and formulate $s_\psi(\tau) = \sum_{h=1}^{H} r_\psi(s_h, a_h)$. During our experiments, we found that both of them can learn an accurate score model in large data regimes (given a large number of preference labels) while the MLP-based model performs better in low-data regimes (given insufficient preference labels). Therefore, we chose the MLP implementation for its robustness.

The key difference between naive improvement training and our preference augmentation is illustrated in Figure 6. The latter samples a pair of trajectories from neighboring blocks, aligning preference gaps. We list the pseudocode in Algorithm 2 and relative hyperparameters in Table 4.

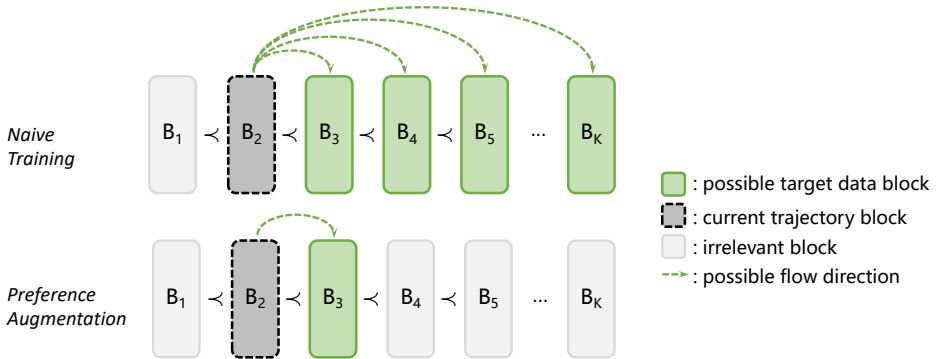

Figure 6: The key difference between the naive training and our preference augmentation.

---

**Algorithm 2** Preference Augmentation

---

**Require:** labeled data $\mathcal{D}^L = \{(\tau_m^1 \succeq \tau_m^0)\}_{m=1}^M$, unlabeled data $\mathcal{D}^U = \{\tau_n\}_{n=1}^N$, block number $K$, batch size $N_{\text{batch}}$.
1: initialize score model $s_\psi$.
2: **for** $i = 1, 2, \cdots$ **do**
3:  $\{(\tau_j^1 \succeq \tau_j^0)\}_{j=1}^{N_{\text{batch}}} \sim \mathcal{D}^L$
4:  $\psi_{i+1} \leftarrow \psi_i - \alpha \nabla_\psi \frac{1}{N_{\text{batch}}} \sum_{j=1}^{N_{\text{batch}}} \left( -\log \frac{\exp s_\psi(\tau_j^1)}{\sum_{k \in \{0,1\}} \exp s_\psi(\tau_j^k)} \right)$
5: **end for**
6: label $\mathcal{D}^U$ by score model $s_\psi$: $\mathcal{D}^U \leftarrow \{\tau_n, s_\psi(\tau_n)\}_{n=1}^N$.
7: cluster $\mathcal{D}^U$ based on scores: $\mathcal{D}^U = B_1 \cup B_2 \cup \cdots \cup B_K$ where $B_1 \prec \cdots \prec B_K$.
**Output:** score model $s_\psi$, blocks $B = \{B_1, B_2, \cdots, B_K\}$.

---

**Algorithm 3** Generative Model Training

---

**Require:** blocks $B_1 \preceq B_2 \preceq \cdots \preceq B_K$, batch size $N_{\text{batch}}$.
1: initialize generative model $p_\theta$.
2: **for** $i = 1, 2, \cdots$ **do**
3:  **for** $k = 1, 2, \cdots, K$ **do**
4:   $\{\tau_j^k\}_{j=1}^{N_{\text{batch}}} \sim B_k$
5:  **end for**
6:  $\theta_{i+1} \leftarrow \theta_i - \alpha \nabla_\theta \frac{1}{K-1} \frac{1}{N_{\text{batch}}} \sum_{k=1}^{K-1} \sum_{j=1}^{N_{\text{batch}}} (-\log p_\theta(\tau_j^{k+1}|\tau_j^k))$   $\triangleright$ $p_\theta$ is calculated by
   Equation 3.
7: **end for**
**Output:** generative model $p_\theta$.

---

### B.3 Generative Model Training

After Preference Augmentation, we can sample pairs of trajectories from neighboring blocks for training a generative model, i.e., trajectory diffuser. We have illustrated its architecture and the training objective in the main text and we now present the detailed pseudocode in Algorithm 3 and the according hyperparameters in Table 5.

Table 4: Hyperparameters of Preference Augmentation.

| Hyperparameter | Value |
|---|---|
| Number of layers | 3 |
| Hidden dimension | 256 |
| Activation | relu |
| Batch size | 256 |
| Learning rate | $1e-4$ |
| Optimizer | Adam |
| Number of blocks $K$ | 20 |
| Cluster method | KMeans |

Table 5: Hyperparameters of Trajectory Diffuser.

| Hyperparameter | Value |
|---|---|
| Guidance scale $\omega$ | 1.2 |
| Diffusion steps $T$ | 1000 |
| Downsample rate | [1, 2, 4, 8] |
| Hidden dimension | 256 |
| Batch size | 64 |
| Dropout | 0.2 |
| Learning rate | $1e-4$ |
| Optimizer | Adam |

Table 6: Hyperparameters of Policy Extraction.

| Task Name | Num trajs for improvement $k$ | Filter threshold $\sigma$ | Weight Decay $\omega$ |
|---|---|---|---|
| D4RL | 300 | 1.05 | 2e-4 |
| Meta-Wolrd | 100 | 1.1 | 1e-3 |
| NeoRL | 300 | 1.05 | 2e-4 |

---

**Algorithm 4** Policy Extraction

**Require:** unlabeled data $\mathcal{D}^U = \{\tau_n\}_{n=1}^N$, score model $s_\psi$, trajectory diffuser $p_\theta$,
        num of trajectories for improvement $k$, filter threshold $\sigma$, flow step $t_{\text{flow}}$, batch size $N_{\text{batch}}$.
1: label $\mathcal{D}^U$ by score model $s_\psi$: $\mathcal{D}^U \leftarrow \{\tau_n, s_\psi(\tau_n)\}_{n=1}^N$.
2: get top-$k$ trajectories $\{\tau_l^0\}_{l=1}^k$ from $\mathcal{D}^U$.
3: $\mathcal{D}^{\text{Improved}} \leftarrow \{\tau_l^0\}_{l=1}^k$.
4: **for** $i = 1, 2, \cdots, t_{\text{flow}}$ **do**
5:     **for** $l = 1, 2, \cdots, k$ **do**
6:         $\tau_l^i \sim p_\theta(\tau_l^{i-1})$         ▷ sample a trajectory from trajectory-diffuser.
7:         **if** $s_\psi(\tau_l^i)/s_\psi(\tau_l^{i-1}) > \sigma$ **then**
8:             $\mathcal{D}^{\text{Improved}} \leftarrow \mathcal{D}^{\text{Improved}} \cup \{\tau_l^i\}$         ▷ add a trajectory only when it is improved.
9:         **else**
10:             $\tau_l^i = \tau_l^{i-1}$
11:         **end if**
12:     **end for**
13: **end for**
14: $\mathcal{D}^{\text{Improved}} \leftarrow$ top $k$ trajectories of $\mathcal{D}^{\text{Improved}}$.
15: Initialize policy $\pi_\phi$.
16: **for** $i = 1, 2, \cdots$ **do**
17:     $\{(s_j, a_j)\}_{j=1}^{N_{\text{batch}}} \sim \mathcal{D}^{\text{Improved}}$
18:     $\phi_{i+1} \leftarrow \phi_i - \alpha\nabla_\phi \frac{1}{N_{\text{batch}}} \sum_{j=1}^{N_{\text{batch}}} (\pi_\phi(s_j) - a_j)^2$     ▷ behavioral cloning.
19: **end for**
**Output:** policy $\pi_\phi$.

---

### B.4 POLICY EXTRACTION

By means of the score model $s_\psi$ and the trajectory diffuser $p_\theta$, we can perform iterative refinements beginning with the top $k$ trajectories in the dataset, in which the number of refinements (flow step) is adaptive. Assuming the block number is $K$ and the trajectory with the lowest score among the top $k$ trajectories is in the $i$-th block, then the flow step $t_{\text{flow}} = K - i$. During generation, it samples perturbed noise $\bar{\epsilon}_\theta(\tau_t^1, t) := \epsilon_\theta(\tau_t^1, \varnothing, t) + \omega(\epsilon_\theta(\tau_t^1, \tau^0, t) - \epsilon_\theta(\tau_t^1, \varnothing, t))$. Here, $\tau^1$ is the sampled target trajectory, $\tau^0$ is the conditional trajectory, and $\omega$ serves as a hyperparameter to adjust the degree of conditional guidance. After that, we can obtain a large number of improved trajectories, from which we can derive a desirable policy via imitation learning. For simplicity, we choose

behavioral cloning for policy extraction, which is implemented by a simple MLP network (a 2-layered MLP with 256 hidden units and ReLU activations). Algorithm 4 presents the full procedure and detailed hyperparameters are listed in Table 6.

### B.5 Data Augmentation for Offline RL

When FTB is used for data augmentation for offline RL, it can be simplified since there is no need for Preference Augmentation. Provided known rewards, we can directly cluster the trajectories into $K$ blocks based on real returns and then train the trajectory diffuser. Note that in this case, we also need to generate rewards in addition to states and actions. After that, we select the top 100 trajectories in the dataset according to returns and iteratively refine them by applying the flow-to-better process ($t_{flow} = 5$). Finally, we take the trajectories in the final flow step of generation as supplement data.

For offline RL training, we follow the default hyperparameters of TD3+BC (Fujimoto & Gu, 2021) and IQL (Kostrikov et al., 2022). For TD3+BC, we set the BC constraint to 2.5. For IQL, we use $\tau = 0.7$ (expectile weight) and $\beta = 3.0$ (inverse temperature).

## C Omitted Experiments

### C.1 Ablation study on block number

The parameter block number $K$ determines the dataset's partition into $K$ blocks in ascending order. We conduct ablation experiments on block number K for two different tasks. The experimental results are shown below, from which it can be observed that our method exhibits robustness to the block number $K$ variations.

Table 7: Ablation study of Block Number $K$.

|  | $K = 10$ | $K = 20$ | $K = 30$ |
|---|---|---|---|
| hopper-medium-replay-v2 | 74.4±2.9 | 89.6±4.9 | 92.8±0.8 |

|  | $K = 3$ | $K = 5$ | $K = 10$ |
|---|---|---|---|
| plate-slide-v2 | 0.44±0.08 | 0.51±0.08 | 0.52±0.04 |

### C.2 Ablation study on architecture

Additionally, we perform ablation experiments to evaluate the influence of the diffusion model's structure on the experimental outcomes. The results suggest that the model structure indeed affects the performance. In cases where the model's expressive capacity is inadequate, the generated trajectories may lack realism, consequently impacting the overall performance.

Table 8: Ablation study of Architecture of Trajectory Diffuser.

|  | **FTB** | **shallower FTB** | **Narrower FTB** |
|---|---|---|---|
| hopper-medium-replay-v2 | 89.6±4.9 | 86.9±2.0 | 88.7±3.8 |
| plate-slide-v2 | 0.51±0.08 | 0.41±0.21 | 0.43±0.05 |

### C.3 Ablation study on dataset distribution

Our method involves block ranking of the dataset, hence requiring a broad distribution of data. To assess the sensitivity of our method to dataset distribution, we conducted additional experiments on the NeoRL benchmark (Qin et al., 2022), which takes into account the above reality gap and has the narrower distribution of data in locomotion tasks. The results of this experiment are shown in Table 9. Different datasets are sampled based on policies of varying quality. We observe that FTB does exhibit performance degradation on tasks with narrower dataset distributions, however, it still outperforms existing baselines.

Table 9: Performance with preference from narrow data distribution, averaged over 5 random seeds. Among the offline PbRL methods, we bold the highest score for each task. Among all the methods, we mark the highest score with "*" for each task.

| Task Name | BC | TD3+BC | IQL | IQL+$r_\psi$ | FTB (Ours) |
|---|---|---|---|---|---|
| Hopper-v3-L-999 | 15.1 | 15.8 | 16.3 | 17.1 | **17.6 ± 1.2**$^*$ |
| Hopper-v3-M-999 | 51.3 | 70.3 | 47.7 | 30.9 | **67.4± 23.4**$^*$ |
| Hopper-v3-H-999 | 43.1 | 75.3$^*$ | 24.8 | 42.6 | 49.7±21.8 |
| **Average** | 36.5 | 53.8 | 29.6 | 30.2 | 44.9 |

## C.4 EXPERIMENTAL RESULTS WITH HUMAN TEACHER

In line with a variety of previous work, we also evaluate FTB with human preference labels. For the sake of fairness, we use the preference labels from PT (Kim et al., 2023), which is available at https://github.com/csmile-1006/PreferenceTransformer/tree/main/human_label. As shown in Table 10, we find that FTB also performs better than previous offline PbRL algorithms.

Table 10: Performance with preference from human teachers, averaged over 5 random seeds. Among the offline PbRL methods, we bold the highest score for each task. Among all the methods, we mark the highest score with "*" for each task.

| Task Name | 10%BC | TD3+BC | IQL | IQL+$r_\psi$ | PT | OPRL | IPL | FTB (Ours) |
|---|---|---|---|---|---|---|---|---|
| halfCheetah-medium-v2 | 42.5 | 48.1 | 48.3 | 46.7 | **48.4**$^*$ | **47.5** | **47.3** | 35.1± 4.7 |
| hopper-medium-v2 | 55.5 | 60.4 | 67.5$^*$ | **64.7** | 38.1 | 59.8 | 50.8 | 61.9±3.6 |
| walker2d-medium-v2 | 67.3 | 82.7$^*$ | 80.9 | **80.4** | 66.1 | **80.8** | 79.5 | **79.7±4.1** |
| halfCheetah-medium-replay-v2 | 23.6 | 44.8$^*$ | 44.5 | 43.2 | **44.4** | 42.3 | 42.5 | 39.0±1.0 |
| hopper-medium-replay-v2 | 70.4 | 64.4 | 97.4$^*$ | 11.6 | 84.5 | 72.8 | 73.6 | **90.8±2.6** |
| walker2d-medium-replay-v2 | 54.4 | 85.6$^*$ | 82.2 | 72.1 | 71.2 | 63.2 | 60.0 | **79.9±5.0** |
| halfCheetah-medium-expert-v2 | 90.1 | 90.8 | 94.7$^*$ | 88.8 | 87.5 | 87.7 | 87.0 | **91.3±1.6** |
| hopper-medium-expert-v2 | 111.2 | 101.2 | 107.4 | 57.8 | 69.0 | 81.4 | 74.5 | **110.0±2.3** |
| walker2d-medium-expert-v2 | 108.7 | 110.0 | 111.7$^*$ | 108.3 | **110.1** | **109.6** | 108.5 | **109.1±0.1** |
| **Average** | 69.3 | 76.5 | 81.6$^*$ | 63.7 | 68.8 | 71.7 | 69.3 | **77.4** |

## C.5 OMITTED COMPARISONS OF REWARD LEARNING AND PREFERENCE LEARNING IN SECTION 4.2

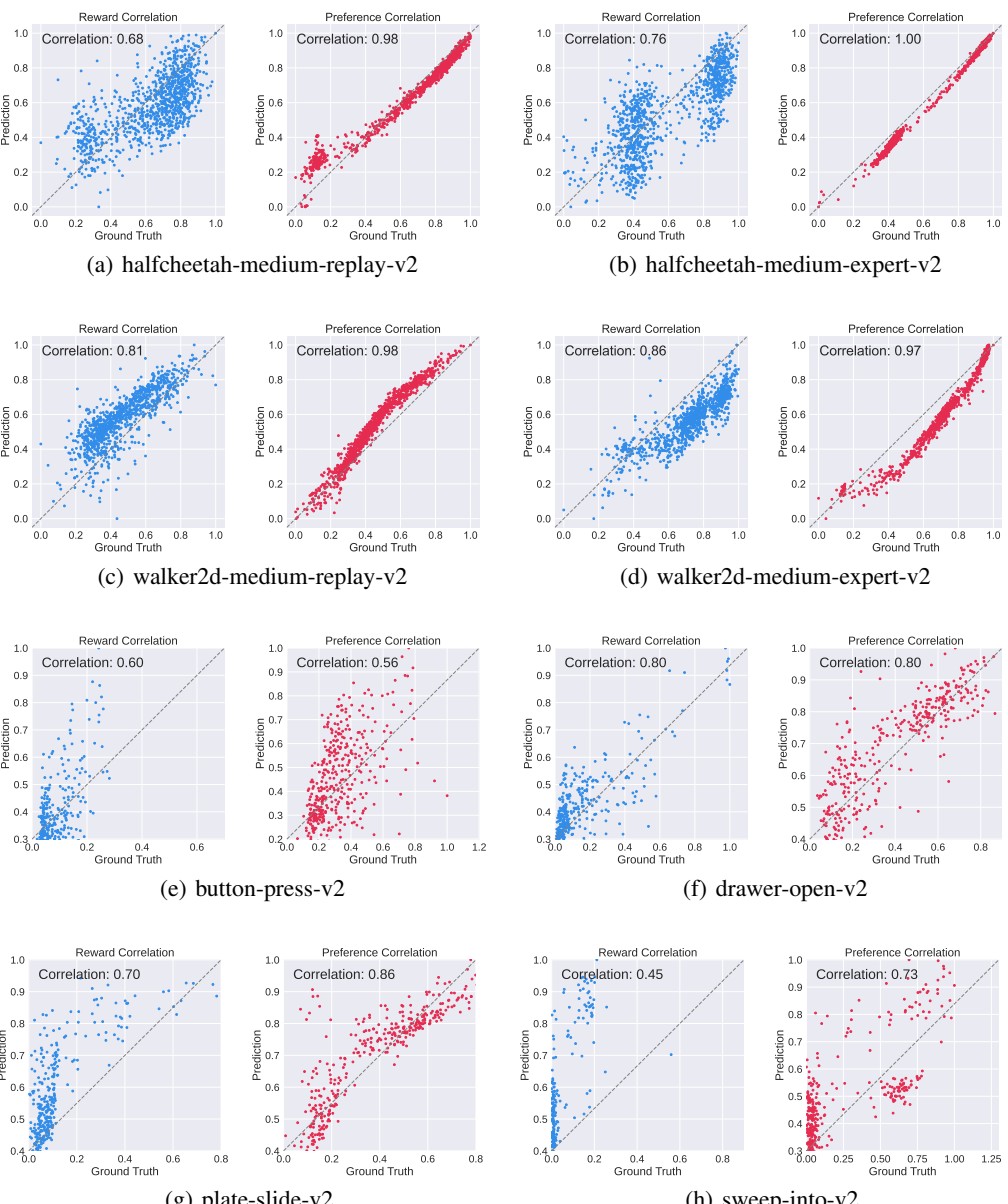

Figure 7: Illustration of the correlation between learned rewards/preferences and the ground-truth rewards/preferences.

## C.6   Omitted generated trajectory analysis in Section 4.3

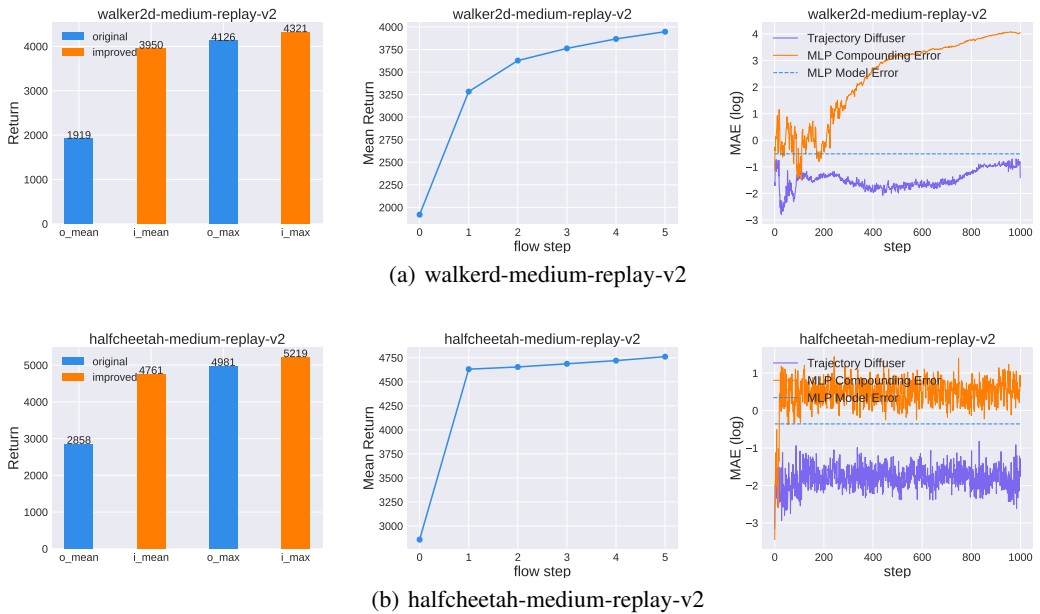

(a) walkerd-medium-replay-v2

(b) halfcheetah-medium-replay-v2

Figure 8: Illustration of performance before and after improvement (left), performance with different flow steps (center), and the dynamics error of generated trajectories (right).

