# OpenReview forum: "Flow to Better: Offline Preference-based Reinforcement Learning via Preferred Trajectory Generation"
_ICLR.cc/2024/Conference — ICLR 2024 poster_

### Official Review · Reviewer_Juf3 · 2023-10-28

**Soundness:** 3 good
**Presentation:** 3 good
**Contribution:** 3 good
**Rating:** 6
**Confidence:** 3

**Summary:**

The paper presents a method called Flow-to-Better (FTB) for offline preference-based reinforcement learning (PbRL). The main objective of offline PbRL is to address the challenges of designing rewards and the high costs of online interactions by providing agents with a fixed dataset containing human preferences between pairs of trajectories. Previous approaches in offline PbRL focus on recovering rewards from preferences and policy optimization using offline RL algorithms. However, accurately learning transition-wise rewards from trajectory-based preference labels can be challenging and lead to misguidance during training. To overcome this challenge, the FTB method leverages the pairwise preference relationship to generate higher-preference trajectories and improve trajectory-level policy behaviors. It uses a diffusion model in a conditional generation process to flow low-preference trajectories to high-preference trajectories. The method also incorporates Preference Augmentation to address the problem of insufficient preference data. The experimental results demonstrate that FTB outperforms previous offline PbRL methods and can serve as an effective data augmentation method for offline RL. The paper provides an introduction to reinforcement learning and discusses the challenges of crafting well-designed rewards. It explains the framework of offline PbRL, provides background information on diffusion models, and details the policy extraction and experimental procedure.

**Strengths:**

This paper introduces a novel method called Flow-to-Better (FTB) for offline preference-based reinforcement learning (PbRL). The paper addresses the problem of learning from preference feedback, where the goal is to learn a policy that maximizes a user's preferences. The key idea behind FTB is to iteratively refine a set of trajectories using a diffusion model, generating new trajectories that exhibit higher preferences.

In terms of originality, FTB introduces a unique approach to offline PbRL by leveraging diffusion models. While previous work has explored the use of generative models for planning, FTB stands out by using the generative model to improve the quality of trajectories without the need for resource-intensive inference. This approach allows for end-to-end trajectory improvement and provides a new perspective on leveraging generative models in offline RL.

The quality of the paper is high, as it presents a well-defined problem formulation, a clear description of the proposed method, and thorough experimental evaluations. The authors provide detailed explanations of the different components of FTB, including the preference augmentation, trajectory diffuser, and policy extraction. The experiments demonstrate the effectiveness of FTB in various continuous control tasks, comparing it with state-of-the-art offline RL methods. The results show significant improvements in performance, highlighting the quality of the proposed approach.

The clarity of the paper is commendable. The authors provide clear definitions of key concepts and algorithms, making it easy to understand the proposed method. The paper is well-structured, with a logical flow of ideas and clear explanations of the experimental setup and results. The use of algorithms and figures further enhances the clarity of the presentation.

In terms of significance, the paper makes several contributions. Firstly, it introduces a new method, FTB, for offline PbRL that leverages diffusion models to improve the quality of trajectories. This provides a valuable approach for learning from preference feedback in RL settings. Secondly, the paper demonstrates the effectiveness of FTB through extensive experiments on continuous control tasks, showing significant improvements over state-of-the-art methods. This highlights the practical significance of the proposed approach.

**Weaknesses:**

One potential weakness of the paper is the lack of a detailed comparison with existing methods in the field of offline preference-based reinforcement learning (PbRL). While the paper provides comparisons with state-of-the-art offline RL methods, it would be beneficial to include a more comprehensive comparison with existing PbRL methods. This would help to establish the novelty and superiority of the proposed FTB method in the specific context of preference-based learning.

Additionally, the paper could benefit from a more thorough discussion of the limitations and potential drawbacks of the proposed method. While the experimental results demonstrate the effectiveness of FTB in improving policy performance, it would be valuable to discuss scenarios or settings where FTB may not perform as well or potential challenges that may arise in its application. This would provide a more balanced perspective and help readers understand the practical limitations of the proposed approach.

Furthermore, the paper could provide more insights into the computational complexity and scalability of the FTB method. Since diffusion models can be computationally expensive, it would be helpful to discuss the computational requirements of the proposed method and any potential trade-offs between performance and computational efficiency. This would provide a clearer understanding of the practical feasibility of applying FTB to large-scale RL problems.

Lastly, the paper could benefit from a more detailed explanation of the hyperparameter choices and their impact on the results. While the paper mentions some hyperparameters, such as the number of flows and the BC constraint, it would be helpful to provide more insights into how these choices were made and their effects on the performance. This would allow readers to better understand the sensitivity of the method to different hyperparameter settings and potentially explore alternative choices in their own applications.

**Questions:**

1. Could you provide more insights into the computational complexity of FTB?

2. Are there any specific scenarios or environments where FTB may not perform well? Providing a more comprehensive discussion on the limitations of FTB would help in understanding its practical applicability and potential areas for future improvement.

3. What are the key factors or design choices in FTB that contribute to its improved performance?

---

> ### Author Response · Authors · 2023-11-18
> **Response to Juf3 (Part 1)**
>
> We thank the reviewer for the valuable comments and time dedicated to evaluating our work.
>
> **Comment 1:** One potential weakness of the paper is the lack of a detailed comparison with existing methods in the field of offline preference-based reinforcement learning (PbRL).
>
> **Response 1:** As suggested, we have included a state-of-art baseline OPRL in Table 1.
>
> |  | OPPO | IQL+MR   | IQL+PT    | OPRL (IQL based) | FTB       |
> | --- | --- | --- | --- | --- | --- |
> | halfcheetah-med-exp | 46.3 | 88.8     | 87.5      | 87.7             | **91.6**  |
> | hopper-med-exp      | 30.6 | 57.8     | 69.0      | 81.4             | **111.8** |
> | walker2d-med-exp    | 89.3 | 108.3    | **110.1** | 109.6            | 109.2     |
> | halfcheetah-med     | 42.5 | 46.7     | **48.4**  | 47.5             | 37.4      |
> | hopper-med          | 45.6 | **64.7** | 38.1      | 59.8             | 61.4      |
> | walker2d-med        | 70.2 | 80.4     | 66.1      | **80.8**         | 78.5      |
> | halfcheetah-med-rep | 30.4 | 43.2     | **44.4**  | 42.3             | 39.3      |
> | hopper-med-rep      | 47.8 | 11.6     | 84.5      | 72.8             | **90.2**  |
> | walker2d-med-rep    | 29.9 | 72.1     | 71.2      | 63.2             | **79.9**  |
> | Average             | 48.1 | 63.7     | 68.8      | 71.7             | **77.4**  |
>
> ---
>
> **Comment 2:** The paper could benefit from a more thorough discussion of the limitations and potential drawbacks of the proposed method.
>
> **Response 2:** Thanks for your constructive suggestion! Yes, I think there are two specific scenarios or environments where FTB may not perform well:
>
> 1. **The dataset has a narrow distribution and poor quality:** The "flow to better" paradigm relies on having several trajectories of varying quality in the dataset to extract hierarchical improvement paths. As a result, FTB may struggle with datasets that have a narrow distribution of trajectories or only consist of random trajectories. Nevertheless, in the context of our application, most scenarios in offline PbRL adhere to the conditions required by our method.
> 2. **The task horizon is excessively long:** Compared to the original methods, FTB has a higher demand for computational resources. The computation time of the attention-based diffusion model grows quadratically with the task's horizon. Therefore, when the task horizon is excessively long, the training duration of FTB will significantly increase. However, we believe this issue might be effectively mitigated through existing solutions such as latent diffusion models.
>
> ---
>
> **Comment 3:** The paper could provide more insights into the computational complexity and scalability of the FTB method.
>
> **Response 3:** We appreciate your valuable suggestions! The diffusion model indeed demands more computational resources compared to the original methods. We conducted a thorough comparison of resource consumption in both Mujoco and MetaWorld tasks, with the specific results outlined above. To ensure fairness, all computations were performed on a RTX 4090 and implemented using the PyTorch framework. Diverging from other planning approaches based on the diffusion model, FTB employs behavior cloning to extract policies from trajectories generated by the diffusion model into an MLP network. This strategic choice enables FTB to enjoy the same advantages in terms of memory usage and inference speed as existing methods during the inference phase. Consequently, this design contributes to the demonstrated superiority of our approach, particularly in larger-scale offline pbrl tasks.
>
> | Training Time | IQL/PT/OPRL | FTB(ours) | GPU Memory | IQL/PT/OPRL | FTB(ours) |
> | --- | --- | --- | --- | --- | --- |
> | Mujoco | 3 hours | 36 hours | Mujoco | 1 Gb | 23 Gb |
> | MetaWorld | 3 hours | 15 hours | MetaWorld | 1.5 Gb | 8 Gb |
>
> ---

---

> > ### Author Response · Authors · 2023-11-18
> > **Response to Juf3 (Part 2)**
> >
> > **Comment 4:** The paper could benefit from a more detailed explanation of the hyperparameter choices and their impact on the results.
> >
> > **Response 4:** Thanks for your valuable suggestion! We have further conducted ablation studies and given more detailed explanations on several hyperparameters including block numbers K and the architecture of the trajectory diffuser.
> >
> > 1. **Block numbers K:** The parameter block number 'K' determines the dataset's partition into 'K' blocks in ascending order. Intuitively, we prefer a larger 'K' since it can avoid the performance gap of trajectories between adjacent blocks being so different that it is challenging for the diffusion model to learn a “worse-to-better” mapping. However, a larger ‘K’ also results in a smaller number of trajectories in each block, which may decrease the diversity of generated trajectories. Therefore, we need to make a trade-off and select a proper ‘K’. Nevertheless, through the analysis of generalization in the original experiment 4.4 and additional ablation experiments on both MetaWorld and Mujoco, we find that our method is robust to the hyperparameter 'K', effectively striking a balance between diversity and quality.
> >
> >
> >     |  | K=10 | K=20 | K=30 |
> >     | --- | --- | --- | --- |
> >     | hopper-medium-replay-v2 | 89.0±1.0 | 90.8±2.6 | 89.3±5.7 |
> >     |  | K=3 | K=5 | K=10 |
> >     | plate-slide-v2 | 0.44±0.08 | 0.51±0.08 | 0.52±0.04 |
> > 2. **Architecture of trajectory diffuser**: Additionally, we performed ablation experiments to evaluate the influence of the diffusion model's structure on the experimental outcomes. The results suggest that the model structure indeed affects the results. In cases where the model's expressive capacity is inadequate, the generated trajectories may lack realism, consequently impacting the overall performance.
> >
> >
> >     |  | Original FTB,                             dim_mutls=[1, 2, 4, 8], hidden_dim=128 | FTB with a shallower depth,                             dim_mutls=[1, 2, 4], hidden_dim=128 | FTB with a narrower width,                             dim_mutls=[1, 2, 4, 8], hidden_dim=64 |
> >     | --- | --- | --- | --- |
> >     | hopper-medium-replay-v2 | 90.8±2.6 | 46.0±1.9 | 70.5±3.3 |
> >     | plate-slide-v2 | 0.51±0.08 | 0.41±0.21 | 0.43±0.05 |
> >
> > ---
> >
> > **Question 1:** Could you provide more insights into the computational complexity of FTB?
> >
> > **Answer 1:**  Please refer to response to comment 3.
> >
> > ---
> >
> > **Question 2:** Are there any specific scenarios or environments where FTB may not perform well? Providing a more comprehensive discussion on the limitations of FTB would help in understanding its practical applicability and potential areas for future improvement.
> >
> > **Answer 2:** Please refer to response to comment 2.
> >
> > ---
> >
> > **Question 3:** What are the key factors or design choices in FTB that contribute to its improved performance?
> >
> > **Answer 3:** We argue that the key factor that makes our method outperform other PbRL baselines is that we avoid using inaccurate learned rewards for TD learning. Instead, we attempt to directly derive a policy through BC from enough high-preference trajectories. To generate more high-preference trajectories, we leverage the powerful expressive capacity of the diffusion model to learn the process of transitioning from “worse” to “better”. Through iteratively improving trajectories in the dataset, we can generate a batch of diverse and more preferred trajectories as augmentation.

---

> ### Author Response · Authors · 2023-11-21
> **Looking forward to your feedback**
>
> Dear Reviewer Juf3:
>
> We first would like to thank the reviewer's efforts and time in reviewing our work. We were wondering if our responses have resolved your concerns. Since the discussion period is ending soon, we would like to express our sincere gratitude if you could check our reply to your comments. We will be happy to have further discussions with you if there are still some remaining questions! We sincerely look forward to your kind reply!
>
> Best regards,
>
> The authors

---

### Official Review · Reviewer_ssfF · 2023-10-31

**Soundness:** 2 fair
**Presentation:** 2 fair
**Contribution:** 2 fair
**Rating:** 5
**Confidence:** 5

**Summary:**

This paper aims to solve the offline preference-based reinforcement learning issue. Different with prior works, this paper proposes a method named Flow-to-Better, which attempts to adopt the pairwise preference relationship to guide a generative model. Then, this work  introduces Preference Augmentation to alleviate the issue of insufficient preference labels. Further, this work explain how to derive a deployable policy and provide a full procedure. The authors conduct experiments in D4RL to verify the effectiveness of their method.

**Strengths:**

1. This paper written well and easy to follow. The structure of this paper is very clear.
2. This paper provide a new solution for OPBRL, that is use the less preferred trajectory to generate high preferred trajectory. Further, this work proposes two techinique to combine with the generative model method: (1) preference augmentation and (2) imitation-based policy extraction.

**Weaknesses:**

1. Although the authors tested their method, an important baseline is missing [1]. This work gives a ensemble-diversed-based OPBRL method. I ran this method on D4RL with IQL and was surprised to find that the effect was very good. It only required less than 5 queries to perform similarly to Oracle.

2. I have serious concerns about the author's experimental results. The correlation coefficient between the reward function in D4RL and the dimension of the observed speed in observation space is very high, so the OPBRL method only needs very few queries to perform very well on D4RL, and the method proposed by the author The superior performance of the method is difficult to measure on D4RL.

[1] Shin, Daniel, Anca D. Dragan, and Daniel S. Brown. "Benchmarks and algorithms for offline preference-based reward learning." arXiv preprint arXiv:2301.01392 (2023).

**Questions:**

Based on the above weakness, I have the following questions:

1. Can you compare with your method with [1] based on IQL in D4RL?

2. Can you provide the correlation ratio between rewards and each dimension in observations in D4RL?

---

> ### Author Response · Authors · 2023-11-18
> **Response to ssfF**
>
> We thank the reviewer for the valuable comments and time dedicated to evaluating our work.
>
> **Comment 1:** Although the authors tested their method, an important baseline is missing [1]. This work gives an ensemble-diversed-based OPBRL method which only required less than 5 queries to perform similarly to Oracle.
>
> **Response 1:** We thank the reviewer for pointing out the related work [1] and we have compared it as shown in responses to comment 2 and question 1. However, we have to point out that OPRL and our setting are not quite the same. Specifically, OPRL allows the agent to continue asking the expert for preference labels during training, while our setting in the paper does not permit this.
>
> ---
>
> **Comment 2:** The superior performance of the method is difficult to measure on D4RL because reward learning in mujoco is relative simple.
>
> **Response 2:** We indeed agree that reward learning in mujoco is relatively simple. To assess the efficacy of FTB in more challenging tasks, we conducted supplementary experiments on the MetaWorld benchmark, which has much more complex reward functions. Our experimental approach closely aligns with that of [1], involving the collection of 400 trajectories for each task through four distinct policy levels. However, we diverged by incorporating only 30 preference pairs, a notable reduction from the 500 pairs employed in [1].
>
> As shown in the following table, learning the reward function for MetaWorld poses challenges with just 30 sample pairs. In this context, 'accuracy' refers to the reward or preference models' ability to accurately determine the partial order relationships within state-action pairs or trajectories. In addition, 'correlation' measures how well the learned rewards (or preferences) align with the actual, ground-truth rewards (or preferences) across the entire dataset. In contrast, preference learning surpasses reward learning in terms of both accuracy and relevance.
>
> | Accuracy | Reward Learning | Preference Learning | Correlation | Reward Learning | Preference Learning |
> | --- | --- | --- | --- | --- | --- |
> | assembly-v2 | 0.7950 | **0.8430** | assembly-v2 | 0.65 | **0.87** |
> | button-press-v2 | 0.7632 | **0.8350** | button-press-v2 | **0.60** | 0.56 |
> | drawer-open-v2 | **0.6996** | 0.6910 | drawer-open-v2 | **0.80** | **0.80** |
> | plate-slide-v2 | 0.8304 | **0.8870** | plate-slide-v2 | 0.70 | **0.86** |
> | sweep-into-v2 | 0.6481 | **0.7620** | sweep-into-v2 | 0.45 | **0.73** |
> | Average | 0.7473 | **0.8036** | Average | 0.64 | **0.76** |
>
> Furthermore, we evaluate the performance of baselines including MR+IQL, PT+IQL, OPPO, OPRL, and FTB on these more challenging tasks and present the results in the following table. We can observe that FTB exhibits superior overall performance on MetaWorld tasks compared to these existing baselines.
>
> |  | OPPO | MR+IQL | PT+IQL | OPRL(IQL) | FTB |
> | --- | --- | --- | --- | --- | --- |
> | assembly-v2 | 0.00±0.00 | 0.00±0.00 | **0.04±0.05** | 0.00±0.00 | 0.02±0.00 |
> | button-press-v2 | 0.23±0.12 | 0.16±0.14 | 0.30±0.09 | 0.57±0.23 | **1.00±0.00** |
> | drawer-open-v2 | 0.26±0.09 | 0.85±0.15 | 0.72±0.24 | 0.63±0.20 | **1.00±0.00** |
> | plate-slide-v2 | 0.00±0.00 | 0.45±0.17 | 0.45±0.15 | 0.44±0.13 | **0.51±0.08** |
> | sweep-into-v2 | 0.00±0.00 | 0.18±0.15 | 0.35±0.30 | 0.21±0.12 | **0.97+0.02** |
> | Average | 0.10 | 0.33 | 0.37 | 0.37 | **0.70** |
>
> ---
>
> ---
>
> **Question 1:** Can you compare your method with [1] based on IQL in D4RL?
>
> **Answer 1:** We appreciate the reviewer for pointing out the insufficiency in our experiments. We include the results of OPRL with IQL in D4RL as follows. The experimental results indicate that employing ensemble-diversed-based rewards in IQL does lead to performance improvements. However, FTB still outperforms OPRL and existing baselines in D4RL.
>
> |  | OPPO | IQL+MR | IQL+PT | OPRL (IQL based) | FTB |
> | --- | --- | --- | --- | --- | --- |
> | halfcheetah-med-exp | 46.3 | 88.8 | 87.5 | 87.7 | **91.6** |
> | hopper-med-exp | 30.6 | 57.8 | 69.0 | 81.4 | **111.8** |
> | walker2d-med-exp | 89.3 | 108.3 | **110.1** | 109.6 | 109.2 |
> | halfcheetah-med | 42.5 | 46.7 | **48.4** | 47.5 | 37.4 |
> | hopper-med | 45.6 | **64.7** | 38.1 | 59.8 | 61.4 |
> | walker2d-med | 70.2 | 80.4 | 66.1 | **80.8** | 78.5 |
> | halfcheetah-med-rep | 30.4 | 43.2 | **44.4** | 42.3 | 39.3 |
> | hopper-med-rep | 47.8 | 11.6 | 84.5 | 72.8 | **90.2** |
> | walker2d-med-rep | 29.9 | 72.1 | 71.2 | 63.2 | **79.9** |
> | Average | 48.1 | 63.7 | 68.8 | 71.7 | **77.4** |
>
> ---
>
> ---
>
> **Question 2:** Can you provide the correlation ratio between rewards and each dimension in observations in D4RL?
>
> **Answer 2:**  As suggested, we plotted the correlation between the dimensions of observations and the reward, and the results can be seen at [https://anonymous.4open.science/r/ICLR2024-7244-4A6F/README.md](https://anonymous.4open.science/r/ICLR2024-7244-4A6F/README.md).

---

> ### Author Response · Authors · 2023-11-21
> **Looking forward to your feedback**
>
> Dear Reviewer ssfF:
>
> We first would like to thank the reviewer's efforts and time in reviewing our work. We were wondering if our responses have resolved your concerns. Since the discussion period is ending soon, we would like to express our sincere gratitude if you could check our reply to your comments. We will be happy to have further discussions with you if there are still some remaining questions! We sincerely look forward to your kind reply!
>
> Best regards,
>
> The authors

---

### Official Review · Reviewer_UxuZ · 2023-11-01

**Soundness:** 3 good
**Presentation:** 3 good
**Contribution:** 3 good
**Rating:** 6
**Confidence:** 4

**Summary:**

This paper introduces Flow-to-Better (FTB), a novel diffusion-based framework for offline preference-based reinforcement learning (PbRL). FTB optimizes policies at the trajectory level without Temporal Difference (TD) learning under inaccurate learned rewards. The method consists of three main components: Preference Augmentation, Generative Model Training, and Policy Extraction. Preference Augmentation is an innovative technique designed to alleviate the issue of insufficient preference labels in the approach. The generative model training employs a classifier-free diffusion model called Trajectory Diffuser to generate improved full-horizon trajectories conditioned on the less preferred ones. Finally, the policy extraction process applies imitation learning to derive a deployable policy from the generated trajectories.
Experimental results on three continuous control tasks from the D4RL benchmark demonstrate that FTB consistently outperforms previous offline PbRL methods. Furthermore, the authors show that the proposed trajectory diffuser in FTB can also be used as an effective data augmentation method for offline RL approaches.

**Strengths:**

1. The paper introduces a novel diffusion-based framework for offline preference-based reinforcement learning (PbRL) called Flow-to-Better (FTB), which optimizes policies at the trajectory level without Temporal Difference (TD) learning under inaccurate learned rewards. This approach is innovative and has the potential to improve the performance of offline PbRL methods.
2. The authors also show that the proposed trajectory diffuser in FTB can be used as an effective data augmentation method for offline RL approaches, further expanding the applicability of the method.

**Weaknesses:**

1. The paper focuses on just three continuous control tasks from the D4RL benchmark. Especially, these tasks are known to have simple reward functions [1,2,3], which makes reward learning trivial on these tasks. More specifically, the third dimension of the state ($v_x$) have a 0.99 correlation with the true reward regardless of the quality of the dataset. From the experiment, the simple baseline, IQL+$r_\psi$, performs well except for the Hopper task, which makes me doubt whether a fancy preference learning/augmentation module is necessary. I believe it would be beneficial to evaluate the method on other types of tasks or environments to demonstrate the proposed method's generalizability, including Antmaze, Maze2d and Meta-world [4] tasks.

2. The proposed method uses a diffusion model, which can be computationally heavy. It is unclear how much is the computation cost of the proposed method compared with previous ones. Also, it would be beneficial to ablate on the architecture and the hyperparameters (e.g., block numers $K$) to demonstrate the effectiveness of the proposed preference augmentation module.

I am willing to increase my score if these concerns can be addressed.

[1] Shin, Daniel, Anca D. Dragan, and Daniel S. Brown. "Benchmarks and algorithms for offline preference-based reward learning." arXiv preprint arXiv:2301.01392 (2023).
[2] Li, Anqi, et al. "Survival Instinct in Offline Reinforcement Learning." arXiv preprint arXiv:2306.03286 (2023).
[3] Hu, Hao et al. “Unsupervised Behavior Extraction via Random Intent Priors.” arXiv preprint arXiv:2310.18687 (2023).
[4] Yu, Tianhe, et al. "Meta-world: A benchmark and evaluation for multi-task and meta reinforcement learning." Conference on robot learning. PMLR, 2020.

**Questions:**

1. How does the proposed method perform on other tasks or environments like Antmaze, Maze2d and Meta-world?

2. How much is the computation overhead of the proposed method compared with baseline methods?

3. How do the architecture and hyperparameters of the preference augmentation module affect the final performance?

---

> ### Author Response · Authors · 2023-11-18
> **Response to UxuZ (Part 1)**
>
> We thank the reviewer for the valuable comments and time dedicated to evaluating our work.
>
> **Question 1:** How does the proposed method perform on other tasks or environments like Antmaze, Maze2d and Meta-world?
>
> **Answer 1:** Thanks to the reviewer for pointing out the insufficiency in our experiments. Mujoco indeed serves as a benchmark with a relatively simple reward structure. In order to validate the effectiveness of FTB on more challenging tasks, we conducted additional experiments on the MetaWorld benchmark. Our experimental design for MetaWorld closely follows that of [1], where we collected 400 trajectories for each task using four different levels of policies. However, we deviated by utilizing only 30 preference pairs, significantly fewer than the 500 pairs used in [1].
>
> The first table shown below gives the overall performance on MetaWorld tasks of baselines as well as our methods, from which we can see that imperfect reward learning will degrade the performance of baselines which are based on TD learning while FTB exhibits superior overall performance on these tasks.
>
> |  | OPPO | MR+IQL | PT+IQL | OPRL(IQL) | FTB |
> | --- | --- | --- | --- | --- | --- |
> | assembly-v2 | 0.00±0.00 | 0.00±0.00 | **0.04±0.05** | 0.00±0.00 | 0.02±0.00 |
> | button-press-v2 | 0.23±0.12 | 0.16±0.14 | 0.30±0.09 | 0.57±0.23 | **1.00±0.00** |
> | drawer-open-v2 | 0.26±0.09 | 0.85±0.15 | 0.72±0.24 | 0.63±0.20 | **1.00±0.00** |
> | plate-slide-v2 | 0.00±0.00 | 0.45±0.17 | 0.45±0.15 | 0.44±0.13 | **0.51±0.08** |
> | sweep-into-v2 | 0.00±0.00 | 0.18±0.15 | 0.35±0.30 | 0.21±0.12 | **0.97+0.02** |
> | Average | 0.10 | 0.33 | 0.37 | 0.37 | **0.70** |
>
> In addition, we show that reward learning is more challenging on those tasks from MetaWorld. We compare learned rewards and learned preferences in terms of accuracy and correlation. Specifically, accuracy refers to the precision of the model's judgment on the partial order relationship when given a pair of (s, a) or trajectories, and correlation reflects the relevance between learned rewards(preferences) and ground-truth rewards(preferences). As shown in the following table, learned rewards have significantly low accuracy and correlation while learned preferences perform better. It is also worth noting that as shown in Section 4.3, the correlation of tasks in mujoco is 0.935 on average, but here the correlation in MetaWorld is 0.64, suggesting that reward learning in MetaWorld is indeed more challenging. We also show some plotting results in [https://anonymous.4open.science/r/ICLR2024-7244-4A6F/README.md](https://anonymous.4open.science/r/ICLR2024-7244-4A6F/README.md).
>
> | Accuracy | Reward Learning | Preference Learning | Correlation | Reward Learning | Preference Learning |
> | --- | --- | --- | --- | --- | --- |
> | assembly-v2 | 0.7950 | **0.8430** | assembly-v2 | 0.65 | **0.87** |
> | button-press-v2 | 0.7632 | **0.8350** | button-press-v2 | **0.60** | 0.56 |
> | drawer-open-v2 | **0.6996** | 0.6910 | drawer-open-v2 | **0.80** | **0.80** |
> | plate-slide-v2 | 0.8304 | **0.8870** | plate-slide-v2 | 0.70 | **0.86** |
> | sweep-into-v2 | 0.6481 | **0.7620** | sweep-into-v2 | 0.45 | **0.73** |
> | Average | 0.7473 | **0.8036** | Average | 0.64 | **0.76** |
>
> ---
>
> **Question 2:** How much is the computation overhead of the proposed method compared with baseline methods?
>
> **Answer 2:** Thank you for pointing out our omission. Frankly, diffusion models require more computational resources compared to the original methods. We conducted a comparison of resource consumption in Mujoco and MetaWorld tasks, and the specific results are as above (To ensure fairness, all results were computed on a RTX 4090 and implemented using the PyTorch framework). However, unlike other planning methods based on the diffusion model, FTB utilizes behavior cloning to extract a MLP-based policy from trajectories generated by the diffusion model. This allows FTB to have the same advantages in terms of memory usage and inference speed as existing methods during inference. This is also the reason why our approach demonstrates superiority in larger-scale offline pbrl tasks.
>
> | Training Time | IQL/PT/OPRL | FTB(ours) | GPU Memory | IQL/PT/OPRL | FTB(ours) |
> | --- | --- | --- | --- | --- | --- |
> | Mujoco | 3 hours | 36 hours | Mujoco | 1 Gb | 23 Gb |
> | MetaWorld | 3 hours | 15 hours | MetaWorld | 1.5 Gb | 8 Gb |
>
> ---
>
> [1] Joey Hejna, Dorsa Sadigh. “Inverse Preference Learning: Preference-based RL without a Reward Function.” (Neurips’23)

---

> > ### Author Response · Authors · 2023-11-18
> > **Response to UxuZ (Part 2)**
> >
> > **Question 3:** How do the architecture and hyperparameters of the preference augmentation module affect the final performance?
> >
> > **Answer 3:** Following your valuable suggestion, we conducted ablation experiments on block number K for two different tasks. The experimental results are shown below, from which it can be observed that our method exhibits robustness to hyperparameter variations.
> >
> > |  | K=10 | K=20 | K=30 |
> > | --- | --- | --- | --- |
> > | hopper-medium-replay-v2 | 89.0±1.0 | 90.8±2.6 | 89.3±5.7 |
> > |  | K=3 | K=5 | K=10 |
> > | plate-slide-v2 | 0.44±0.08 | 0.51±0.08 | 0.52±0.04 |
> >
> > Furthermore, we conducted ablation experiments to assess the impact of the diffusion model's structure on experimental results (We presume the 'architecture' you refer to in the question is the structure of the diffusion model). The experimental results indicate that the model structure does influence the outcomes. If the model's expressive capacity is insufficient, the generated trajectories may lack realism, subsequently affecting overall performance.
> >
> > |  | Original FTB,                             dim_mutls=[1, 2, 4, 8], hidden_dim=128 | FTB with a shallower depth,                             dim_mutls=[1, 2, 4], hidden_dim=128 | FTB with a narrower width,                             dim_mutls=[1, 2, 4, 8], hidden_dim=64 |
> > | --- | --- | --- | --- |
> > | hopper-medium-replay-v2 | 90.8±2.6 | 46.0±1.9 | 70.5±3.3 |
> > | plate-slide-v2 | 0.51±0.08 | 0.41±0.21 | 0.43±0.05 |
> >
> > ---

---

> ### Author Response · Authors · 2023-11-21
> **Looking forward to your feedback**
>
> Dear Reviewer UxuZ:
>
> We first would like to thank the reviewer's efforts and time in reviewing our work. We were wondering if our responses have resolved your concerns. Since the discussion period is ending soon, we would like to express our sincere gratitude if you could check our reply to your comments. We will be happy to have further discussions with you if there are still some remaining questions! We sincerely look forward to your kind reply!
>
> Best regards,
>
> The authors

---

> ### Comment · Reviewer_UxuZ · 2023-11-22
>
> Thank you for the detailed response. The response addressed my main concerns. The meta-world experiment shows the general applicability of FTB, and the ablation study provides additional insights into the diffusion module. Therefore, I increased my score to 6. Nevertheless, I encourage the authors to conduct more experiments on other environments like Antmaze to further strengthen the paper and explicitly mention the computational cost in the paper.

---

> > ### Author Response · Authors · 2023-11-22
> >
> > We are glad that our response has addressed the most of your concerns, and we commend your effort in evaluating our paper and raising the score. Your valuable suggestions are greatly appreciated. As part of our commitment to improvement, we intend to conduct more experiments in other environments such as Antmaze. Moreover, we will explicitly incorporate details about the computational cost in our paper.

---

### Author Response · Authors · 2023-11-18
**General Response**

We would like to express our sincere gratitude to the reviewers for their valuable comments and suggestions that have greatly helped us improve our work. We are greatly encouraged by the positive comments of reviewers, e.g.,

- This paper is written well and easy to follow. The structure of this paper is very clear.
- This approach is innovative and has the potential to improve the performance of offline PbRL methods.

We have incorporated the reviewers' suggestions by adding extra explanations, experiments, and discussions on additional related works. The significant revisions are summarized as follows:

- **Including a important baseline**: We included additional discussions and experiments to compare with OPRL [2] in Section 4.1, according to the comments of reviewers ssfF and Juf3.
- **Extra experiments on MetaWorld**: We added extra experiments on MetaWorld [1] tasks in Section 4.1, which are more challenging in terms of rewards learning, according to the comments of reviewers UxuZ and ssfF.
- **More ablation experiments**:  We conducted ablation experiments on hyperparameters block number K of the preference augmentation module and the architecture of the trajectory diffuser in Appendix C.3 and C.4, according to the comments of reviewers UxuZ and Juf3.
- **Discussion of computational consumption**: We discussed the computational consumption of FTB and compared it with previous methods in Appendix B.1, according to the comments of reviewers UxuZ and Juf3.

We now individually address the concerns of reviewers. Please see responses below each review.

[1] Yu, Tianhe, et al. "Meta-world: A benchmark and evaluation for multi-task and meta reinforcement learning." Conference on robot learning. PMLR, 2020.

[2] Shin, Daniel, Anca D. Dragan, and Daniel S. Brown. "Benchmarks and algorithms for offline preference-based reward learning." arXiv preprint arXiv:2301.01392 (2023).

---

### Meta-Review · Area_Chair_cTEH · 2023-12-14

**Metareview:**

This paper introduces the Flow-to-Better(FTB) method, which addresses challenges in offline preference-based reinforcement learning. FTB leverages pairwise preference relationships to guide a generative model in producing preferred trajectories, avoiding the need for Temporal Difference learning with inaccurate rewards. The paper demonstrates the method's effectiveness through multiple experiments, showcasing superior performance compared to existing baselines. The use of limited preference data is a concern. There is a lot of room to improve the neural network architecture. Overall, the paper shows promise in improving trajectory generation and reducing learning complexity in offline preference-based reinforcement learning.

**Justification For Why Not Higher Score:**

There are clear weaknesses on the experimental side, e.g. the use of limited preference data, the comparisons over existing benchmarks.

**Justification For Why Not Lower Score:**

Additional experiments and discussions are helpful to keep it above the borderline.

---

### Decision · Program_Chairs · 2024-01-16

Accept (poster)